# Digested Cinnamon (*Cinnamomum verum* J. Presl) Bark Extract Modulates Claudin-2 Gene Expression and Protein Levels under TNFα/IL-1β Inflammatory Stimulus

**DOI:** 10.3390/ijms24119201

**Published:** 2023-05-24

**Authors:** Elena Lonati, Gessica Sala, Paolo Corbetta, Stefania Pagliari, Emanuela Cazzaniga, Laura Botto, Pierangela Rovellini, Ilaria Bruni, Paola Palestini, Alessandra Bulbarelli

**Affiliations:** 1School of Medicine and Surgery, University of Milano-Bicocca, Via Cadore 48, 20900 Monza, Italy; 2Bicocca cEnter of Science and Technology for FOOD (BEST4FOOD), University of Milano-Bicocca, Piazza della Scienza 2, 20126 Milan, Italy; 3Milan Center for Neuroscience (NeuroMI), School of Medicine and Surgery, University of Milano-Bicocca, Via Cadore 48, 20900 Monza, Italy; 4ZooPlantLab, Department of Biotechnology and Biosciences, University of Milano-Bicocca, Piazza della Scienza 2, 20126 Milan, Italy; 5Innovhub Stazioni Sperimentali per l’Industria S.r.l., Via Giuseppe Colombo 79, 20133 Milan, Italy

**Keywords:** inflammation, IBD, intestinal barrier, cinnamon extract, polyphenols, in vitro digestion, claudin-2, autophagy

## Abstract

Epigenetic changes, host–gut microbiota interactions, and environmental factors contribute to inflammatory bowel disease (IBD) onset and progression. A healthy lifestyle may help to slow down the chronic or remitting/relapsing intestinal tract inflammation characteristic of IBD. In this scenario, the employment of a nutritional strategy to prevent the onset or supplement disease therapies included functional food consumption. Its formulation consists of the addition of a phytoextract enriched in bioactive molecules. A good candidate as an ingredient is the Cinnamon verum aqueous extract. Indeed, this extract, subjected to a process of gastrointestinal digestion simulation (INFOGEST), exhibits beneficial antioxidant and anti-inflammatory properties in an in vitro model of the inflamed intestinal barrier. Here, we deepen the study of the mechanisms related to the effect of digested cinnamon extract pre-treatment, showing a correlation between transepithelial electrical resistance (TEER) decrement and alterations in claudin-2 expression under Tumor necrosis factor-α/Interleukin-1β (TNF-α/IL-1) β cytokine administration. Our results show that pre-treatment with cinnamon extract prevents TEER loss by claudin-2 protein level regulation, influencing both gene transcription and autophagy-mediated degradation. Hence, cinnamon polyphenols and their metabolites probably work as mediators in gene regulation and receptor/pathway activation, leading to an adaptive response against renewed insults.

## 1. Introduction

Inflammatory bowel disease (IBD) is characterised by chronic or remitting/relapsing intestinal tract inflammation. Ulcerative colitis (UC) and Crohn’s disease (CD) are the most diffused forms, with an increasing incidence worldwide [1,2]. Moreover, recent epidemiological data has highlighted the growing number of IBD patients among the elderly, discovering the phenomenon of inflammaging, a state of chronic inflammation driven by cellular senescence and strictly related to biological ageing [3]. The aetiology of IBD is still unclear, however, and different causes are assigned to the disease onset and progression: epigenetic changes and host–gut microbiota interactions along with well-known genetic and environmental factors [4]. Indeed, lifestyle, in terms of physical activity and diet, may contribute to IBD onset. It has been demonstrated that reduced physical activity is associated with a higher risk of CD development [5]. In addition, a diet characterised by a high carbohydrate and red meat intake associated with a low dietary fibre intake, typical of the Western population, negatively influences the microbiota diversity leading to its depletion [4]. On the other hand, a high intake of fibre and tea is important for preventing CD and UC development [6]. Nowadays, several pharmacological therapies are used alone or in combination, but the treatment fails in many patients, or worse, it can lead to rare and serious adverse effects [2]. For these reasons, several emerging studies are oriented toward developing nutritional strategies that include functional food consumption as preventive or supplementary treatment in chronic diseases [7]. Indeed, ingredients in functional foods are different bioactive molecules exerting one or more pleiotropic effects, such as antioxidant, anti-inflammatory, hypolipidemic, glycaemic regulating, cytoprotective, and neuroprotective effects [8]. We recently demonstrated that Cinnamon verum extract, obtained through aqueous extraction, and subjected to a process of gastrointestinal digestion simulation (INFOGEST), retains several beneficial antioxidant and anti-inflammatory functions. Our results showed that polyphenols and other bioactive compounds, contained in cinnamon after digestion, could prevent intestinal barrier permeability and the stress pathways induced by inflammation interacting with cellular mechanisms [9]. In particular, we demonstrated the capability of cinnamon pre-treatment to reduce the loss of transepithelial electrical resistance (TEER), induced by pro-inflammatory cytokines, partially saving the barrier integrity. Since the preservation of intestinal barrier integrity is fundamental to avoiding dysbiosis and worsening inflammation, the cinnamon treatment could be helpful in extending the remission phase. Nevertheless, the underlying molecular mechanisms need to be disclosed. Considering that the barrier’s functionality is due to the tight junction (TJ) protein complex, which is scrambled with an increased insertion of claudin-2 protein during inflammation, it is possible to hypothesise a correlation between TEER decrement and alterations in claudin-2 protein levels. Therefore, this study aimed to go into the details of digested-cinnamon-extract–mediated mechanisms involved in the rescue of intestinal barrier integrity under inflammation. Here we demonstrate that cinnamon pre-treatment plays a positive role in regulating claudin-2 levels in relation to TEER values, through a fine modulation of claudin-2 autophagy-mediated degradation and gene transcription balance.

## 2. Results and Discussion

### 2.1. Correlations between TEER and Claudin-2 Expression in Intestinal Barrier Integrity under Inflammatory Stimulus

Inflammatory bowel diseases (IBDs) are characterised by the loss of intestinal epithelial barrier integrity, increased circulating inflammatory mediators, and excessive tissue injury. In the last twenty years, changes in eating habits have negatively influenced IBD incidence. Pharmacological therapies do not always get the desired results, and thus the interest in developing diets including functional food has grown recently. A correct intake of polyphenol-rich phytoextracts used as ingredients of functional food should be an efficient strategy for prevention and/or a longer remission phase. Data published in a previous paper show that pre-treatment with cinnamon extract, subjected to a digestive protocol, is able to prevent, in a dose-dependent manner, the transepithelial electrical resistance (TEER) loss induced by pro-inflammatory cytokine exposure for 24 h [9]. In the literature, it is well documented that Tumor necrosis factor-α (TNF-α) treatment dose-dependently reduces TEER in a Caco-2 human intestinal barrier model [10,11]. Therefore, here we analysed the effect of cinnamon on barrier integrity at a concentration of greater efficacy on TEER rescue (46 µg/mL) as well as at an earlier time point (6 h) in parallel to the 24 h mark. Interestingly, TEER already began to decrease 6 h after the pro-inflammatory treatment but digested cinnamon extract did not exert any initial rescue effect, as shown by the ΔTEER histograms represented in Figure 1. According to the data, cinnamon polyphenols and their metabolites probably act in a second phase leading to an adaptive response against the insult, working as mediators in cell–cell signalling, receptor and pathway activation, and in gene regulation [12]. Considering the strong interplay between the inflammatory mechanism and oxidative stress, we must take into account that the demonstrated antioxidant properties of digested cinnamon extract might participate in the long-term attenuations of acute events. The activation of endogenous antioxidant defence partially blocks the vicious loop that triggers the inflammatory events, diminishing ROS production and the intracellular pro-oxidant events.

It is worth noting that intestinal barrier dysfunctionality and loss of integrity are often associated with modifications of the tight cell–cell junction protein levels (such as claudins, occludin, and zonula occludens-1). These proteins are differently altered in the inflammatory diseases of the gastrointestinal tract, suggesting a specific profile in every pathology; nevertheless, an increase in claudin-2 expression is the common denominator in IBD [13]. Claudin-2 is a pore-forming claudin that has been shown to largely affect TEER through the modulation of ion channels [14,15]. Thus, claudin-2 protein levels were evaluated under our different experimental conditions. According to the TEER measurements, the cinnamon treatment did not significantly alter the claudin-2 protein levels compared to a control condition, while inflammation induced about a 2-fold and 6-fold increase after 6 and 24 h from cytokine exposure, respectively (Figure 2A). Again, at an earlier time point, the digested cinnamon extract did not exert a significant effect on claudin-2 protein levels with respect to inflammatory conditions. On the contrary, the pre-treatment significantly reduced claudin-2 protein levels by about 35% after 24 h of exposure (Figure 2A), confirming the hypothesis of a late effect on the regulatory mechanisms, preserving the barrier integrity.

The strong correlation observed between TEER analysis and claudin-2 levels is shown in Figure 2B. Since claudin-2 protein expression correlates with disease severity [16,17,18], the pre-treatment with digested cinnamon extract might be an efficient preventive approach to regulate this protein boost and the consequent loss of barrier integrity in acute inflammation events. In a commentary, Barrett discussed the role of claudin-2 in the leak pathways of intestinal permeability, considering the effect of protein overexpression or knockout, and concluding that claudin-2 expression/turnover should be finely regulated in order to balance its levels [19]. Indeed, several studies have investigated claudin-2 regulation by analysing different mechanisms, from transcription to degradation.

### 2.2. Autophagy Modulation in Cells Exposed to Cytokine

Claudin-2 is a high-turnover protein, whose recycling is mediated by endocytosis from the tight junction and by lysosomal/autophagy degradation [20,21,22]. Claudin-2 lysosomal degradation through the macroautophagy machinery has been recognized as a protective mechanism against epithelial integrity leak. Hence, the involvement of the autophagy pathway in claudin-2 reduction observed in pre-treated cells at 24 h from inflammation was evaluated. The most widely used indirect method for monitoring autophagic flux activation is through the analysis of LC3-II expression and lysosomal turnover by the use of specific lysosome inhibitors [23,24]. LC3-II is the lipidated form of LC3. The transition from LC3-I to LC3-II occurs upon autophagy induction and correlates to autophagosome accumulation. In parallel, LC3 mRNA increase has been observed often during autophagy induction [25,26]. As an associated parameter, p62 (SQTM1/sequestosome 1) protein level fluctuations are routinely evaluated. Thus, both mRNA and protein levels of LC3 and p62 were analysed here. At 6 h, LC3 mRNA and LC3-II protein were significantly induced by the pro-inflammatory stimulus regardless of the presence of digested cinnamon extract, suggesting an initial increase in autophagosomes mainly induced by cytokine cocktail exposure. At 24 h, the mRNA synthesis decreased even below the control levels (Figure 3A).

Despite the downregulation of the transcription machinery under TNF-α/IL-1β, LC3-II protein levels were unchanged with respect to the early time point (6 h) and maintained significantly higher than in the untreated cells. On the other hand, the cinnamon pre-treatment led to a significant decrement of about 40% in LC3-II with respect to inflamed cells (Figure 3B). Since all the autophagy vacuole (AV) content is degraded, including the LC3-II localised in its inner membrane when autophagosomes fuse with lysosomes [23], we can speculate that the polyphenol-enriched cinnamon extract promotes a faster rate of AVs degradation with respect to what is observed in presence of cytokine exposure alone. Accordingly, several bioactive compounds regulate autophagy activation, including resveratrol, epigallocatechin-3 gallate, curcumin, quercetin, apigenin, luteolin and kaempferol by themselves or by synergistic effect [7,27]. Among the polyphenols identified in the digested cinnamon extract, apigenin has been revealed as a key modulator of autophagy [28]. Indeed, this flavone seems to induce autophagy flux in cells exposed to an inflammatory stimulus, mediating the interplay between oxidative stress, autophagy, and inflammation [29].

In order to confirm this hypothesis and the activation of autophagic flux, cells in every experimental condition were treated in parallel with Bafylomicin A1 (BafA1), the inhibitor of AV–lysosome fusion. As already described, LC3-II protein levels doubled at 6 h in cells treated with cytokines independently of the presence of digested cinnamon extract, confirming a cytokine-induced modulation of the autophagy pathway over short time periods. Furthermore, during the following 18 h, the treatment with BafA1 significantly increased the LC3-II protein levels, about 8-fold (vs. 46μg/mL + TNF-α*/* IL-1β) or 6-fold (vs. TNF-α /IL-1β) (Figure 3C), confirming a faster rate in the presence of digested cinnamon extract. Indeed, prolonged autophagy induction due to inflammatory stimuli could be more harmful than beneficial. Thus, pre-treatment with digested phytoextract might partially counteract an excessive autophagy induction, restoring a steady-state condition, and improving the proteostasis and cellular quality control. It is worth noting that crosstalk between degradative and secretory autophagy might occur with the release of AVs through unconventional secretory routes and the consequent dangerous inflammatory mediator clearance [30]. Furthermore, a recently identified secretory route named SALI (secretory autophagy during lysosome inhibition) might be activated during the first hours of BafA1 treatment [31], with different magnitudes depending on the experimental conditions. Therefore, LC3-II decrement in cinnamon-pre-treated cells might be the result of the activation of both secretory and degradative autophagy [32], a hypothesis deserving of future insights and experiments.

In parallel, p62 protein level analysis was performed to elucidate autophagy activation. Fluctuations in p62 are routinely investigated to monitor the autophagy flux since this cargo protein is mainly involved in selective autophagy and degraded with its substrate in AVs after lysosome fusion [33,34]. As shown in Figure 4A, under inflammatory stimulus p62 mRNA levels doubled at 6 h, with a slight increment when cells were also pre-treated with cinnamon. In the following 18 h, p62 mRNA synthesis slightly decreased, although it was maintained higher than in the untreated cells. Despite expectations related to AV increment, p62 protein levels were doubled at 6 h from cytokine exposure (Figure 4B).

Moreover, at 24 h, p62 levels in TNF-α/IL-1β-treated cells were higher than in the control ones but significantly decreased with respect to the same condition at 6 h, indicating a partial degradation by selective autophagy in a second phase from an inflammatory stimulus. The data were confirmed by BafA1 co-treatment. Indeed, the p62 protein level increased when AV–lysosome fusion was impaired at both time points, with a stronger effect at 24 h, suggesting an increment in autophagic flux magnitude (Figure 4C). It is worth noting that the higher levels of p62 protein might be due to its multiple roles in pathways activated in response to TNF-α/IL-1β treatment. As shown by the evidence of several experimental studies, p62 is not only involved in autophagy degradation but is also a hub for pro- and anti-inflammatory pathways, positively regulating transcription factors, such as Nrf2 and Nf-κB, under stress conditions [35,36]. Indeed, the Nrf2 inhibitor Keap1 is driven to autophagy degradation by the p62 protein releasing the transcription factor that in turn stimulates SQSTM1 transcription [37]. Moreover, under the inflammatory stimulus, Nf-κB participates in p62 transcription in a feed-forward loop. Since after cytokine treatment significant differences are not appreciable between cells, whether pre-treated or not, but an increasing trend of p62 mRNA and protein under cinnamon pre-treatment is observable, we could speculate that p62 might be mainly engaged in these regulatory pathways rather than only in the autophagic machinery. In particular, considering that cinnamon pre-treatment leads to Nf-κB decrease at 24 h and induces diverse anti-oxidant mechanisms [9], p62 might be transcripted by Nrf2 and involved in the intracellular balance between the inflammatory stimulus and survival signalling.

### 2.3. Effect of Cinnamon Extract on Claudin-2 Autophagy-Mediated Degradation

To clarify the effect of cinnamon extract on claudin-2 autophagy-mediated degradation, claudin-2 protein levels were analysed in all experimental conditions and at all time points, also in the presence of the autophagy inhibitor BafA1. Interestingly, as shown in Figure 5, at 6 h, BafA1 treatment did not affect claudin-2 protein levels, which are comparable in each condition with those observed in the absence of the AV-lysosome fusion inhibitor, suggesting that at this time point, autophagy is not involved in claudin-2 protein level regulation.

Instead, at 24 h, BafA1 increased claudin-2 protein levels in all experimental conditions with a marked effect in cells exposed to pro-inflammatory stimuli. The results confirm the role of autophagy in claudin-2 protein level regulation in a second phase of inflammation. Interestingly, as already observed in the absence of BafA1, pre-treatment with digested cinnamon extract led to a 30% reduction in claudin-2 protein levels in inflamed enterocytes. These data are in line with the trend observed for LC3-II protein expression and suggest that pre-treatment with cinnamon extract is able to partially counteract the increase in claudin-2 induced by the pro-inflammatory stimulus through a slightly faster autophagy flux. Nevertheless, autophagy activation is not sufficient to fully explain the observed claudin-2 decrement in pre-treated cells, and thus its transcriptional regulation was also investigated.

### 2.4. Claudin-2 Transcription Modulation in Cells Exposed to Cytokine

Claudin-2 expression is modulated by a multitude of signalling pathways. Several pro-inflammatory cytokines have been shown to regulate different transcription factors, such as AP-1, NF-κB, Cdx, STAT, and others, inducing TJ protein overexpression [38]. On the other hand, treatment with polyphenols down-regulates claudin-2 mRNA synthesis as a protective mechanism against barrier leak [39]. In order to understand the role of transcriptional regulation in the protein increase after TNF-α/IL-1β exposure, claudin-2 mRNA levels were evaluated. In the first 6 h, claudin-2 mRNA increased about 11-fold after cytokine cocktail exposure with respect to untreated cells, while in the following 18 h, its mRNA was significantly reduced by more than half, suggesting a shutdown of the transcriptional machinery (Figure 6).

In tubular cells (LLC-PK1), long-term treatment with TNF-α reduced the CLDN-2 promoter activity, exerting a biphasic regulation of claudin-2 expression which corresponded to its protein levels [40]. By contrast, here we observed a discrepancy between mRNA and protein levels at both time points, suggesting that the significant increment in claudin-2 protein observed at 24 h is probably due to a successive translation of the initially generated mRNA.

In this scenario, digested cinnamon extract pre-treatment alone did not affect the claudin-2 transcription, while cells subjected to inflammatory stimulus cinnamon pre-treatment seemed to maintain lower mRNA levels at both time points (Figure 6). Nevertheless, at 24 h, claudin-2 expression differs more between only inflamed and pre-treated/inflamed enterocytes, suggesting a long-term effect of the cinnamon extract compounds, also acting on regulatory mechanisms against the pro-oxidant stimuli following the TNF-α/IL-1β exposure [12].

Thus, the cinnamon extract effect on claudin-2 protein decrement might be also due to the regulation of mRNA synthesis. Interestingly, it has been demonstrated that coumarin and its derivatives, which are enriched in the cinnamon extract after digestion, play an important anti-inflammatory role in modulating some of the claudin-2 transcription factors [41]. These include NF-κB, whose phosphorylation and activation are down-regulated by the presence of digested cinnamon extract at 24 h [9], suggesting, at least in part, the involvement of NF-κB in claudin-2 transcription decrease. The elevated amounts of trans-cinnamic acid revealed in our digested extract might also play a key role in the modulation of claudin-2 transcription, according to Ruwizhi and colleagues [42]. Finally, the involvement of specific miRNA in claudin-2 expression is also possible. Indeed, several bioactive food components such as vitamins, polyphenols, minerals, and PUFAs exert interesting anti-inflammatory activity by targeting miRNA [43]. Since the involvement of miR-195-5p in the regulation of claudin-2 transcription has been recently demonstrated [44], we might hypothesize a similar mechanism by a digested cinnamon extract that should be deeply evaluated in the future. 

## 3. Materials and Methods

### 3.1. Cinnamon Extract Preparation and Gastrointestinal In Vitro Digestion

Dried cinnamon bark (*C. verum* L., batch number: C1900010, date: 16 January 2019) was provided by EPO s.r.l. and powdered with an electric laboratory grinder. The cinnamon extraction method has been described by De Giani et al. (2022) [45]. After the extraction process from 1 g of powder with hot distilled water in a rotavapor and polysaccharide precipitation, the extract was filtered under vacuum using Whatman filters n.1. Finally, the dry fraction was resuspended in 5–10 mL of water. The samples were freeze-dried and stored at −20 °C. The gastrointestinal digestion simulation was carried out following the INFOGEST protocol described and used in Minekeus et al. (2014) [46]. The method was detailed in a previous paper [9]. Briefly, the gastrointestinal simulation was performed in 3 phases(oral, gastric, and intestinal)by preparing specific mixtures of salts and enzymes and adjusting the pH and temperature. The digestion product was subjected to acidification in order to precipitate the enzymes and preserve the polyphenols according to Pineda-Vadillo et al. (2016) [47]. Finally, protease inhibitors were added to conserve the samples.

### 3.2. Chemical Analysis and Quantification of Polyphenols in Cinnamon Extract before and after Digestion

In order to determine the total content of polyphenols in the cinnamon extract, the Folin–Ciocalteu assay was carried out with some modifications for the extraction and digested products, as previously described in [9]. The choice was due to the liquid physical state of samples and the necessity of avoiding the interference caused by the presence of enzymes and salts used in the digestion process. The indications detailed in Helal and Tagliazucchi (2018) [48] have been followed.

### 3.3. UHPLC-DAD-ESI-HRMS Profile of Cinnamon Extract before and after Digestion

The bioactive molecule profile of cinnamon extract before and after digestion and the methodology utilised for the analyses were reported in a previous paper [9]. Briefly, a Thermo Vanquish UHPLC system coupled with a Thermo Orbitrap Exploris 120 mass spectrometer and a Vanquish Diode Array Detector (Thermo Scientific, Rodano, Italy) was used to identify the target bioactive molecules. The chromatographic separation was carried out on a Luna Omega Polar C18 (150 mm × 2.1 mm, 3 µm) (Phenomenex, Castelmaggiore, Italy). Phenolic compounds were identified based on the corresponding spectral characteristics (UV and MS/MS spectra), accurate molecular mass, characteristic MS fragmentation pattern, and library comparison in a semi-automatic way through Compound Discoverer Software (2.1, Thermo Scientific, Rodano, Italy). A semi-quantification was carried out to reveal the changes in the compound composition of the cinnamon extract before and after digestion. All compounds were quantified in duplicate at 280 nm by external calibration lines, dividing them into two major groups according to their structural similarity: trans-cinnamic acid or catechin. The data are reported in Pagliari et al., 2023 [9].

### 3.4. Caco-2 Cell Cultures and Intestinal Barrier In Vitro Model

The Caco-2 (ATCC^®^ HTB-37™) human colorectal cancer cell line is a continuous line of heterogeneous human epithelial colorectal adenocarcinoma cells. Caco-2 cells differentiate into enterocytes on collagen-coated Transwell^®^ polyester membrane inserts (Corning 3640). Enterocytes are characterised by tight junctions, microvilli, and a number of specific enzymes and transporters [49]. Caco-2 cells were grown in EMEM medium supplemented with 10% heat-inactivated foetal bovine serum (FBS), 2 mM L-glutamine, 0.1 mM non-essential amino acids, 100 U/mL penicillin, and 100 µg/mL streptomycin, and maintained at 37 °C in a humidified environment with 5% CO_2_. Before the experiments, cells were seeded at 90,000 cells/cm^2^ onto collagen-coated Transwell^®^ membrane inserts and maintained in culture for 21 days in order to reach a reliable intestinal barrier. The transepithelial electrical resistance (TEER) was measured every 7 days by means of an EVOM EndOhm-12chamber (World Precision Instruments, Sarasota, FL, USA), as a parameter of barrier attainment. The medium was replaced every 3 days in the basolateral and the apical chambers, which represent the circulatory and luminal poles of the intestinal epithelium, respectively.

### 3.5. Cell Treatments

In order to mimic acute inflammation, Caco-2 cells were treated with a cocktail of pro-inflammatory cytokines (TNF-α 10 ng/mL + IL-1β 5 ng/mL) for 6 or 24 h, by a basolateral administration, as described by Van de Walle et al. (2010) [50] with minor modifications. Digested cinnamon extract was dispensed to the apical side at a concentration of 46 µg/mL in polyphenol content, during the 24 h preceding the inflammatory stimulus. The concentration was selected according to the data collected in a previous study of the antioxidant and anti-inflammatory properties of this compound, while the timing was chosen to mimic a more physiologic chronic exposure to a low polyphenol concentration. To deepen the treatment effect on the autophagy mechanisms, different groups of Caco-2 cells (untreated, inflamed, and pre-treated with cinnamon extract alone or in combination with inflammatory stimulus) were treated with BafA (#54645 Cell Signaling Technology, Danvers, MA, USA), an inhibitor of autophagosome–lysosome fusion. BafA1 were administered after cytokine exposure for 4 h or 18 h at a 100 nM concentration, as suggested by the manufacturer’s instructions and the literature [51]. TEER was measured before and after the 24 h inflammatory treatment for every insert, and the ΔTEER [Δ(Ω.cm^2^)] between the two measures was calculated and analysed within the groups, to evaluate barrier functionality following diverse treatments.

### 3.6. Electrophoresis and Immunoblotting

The samples were analysed by SDS-PAGE electrophoresis, loading equal amounts of homogenate on 12.5% polyacrylamide Tris-glycine gels, and transferring to a nitrocellulose membrane (Amersham, GE Healthcare Europe GmbH, Milano, Italy). Specific antibodies were employed to reveal protein levels by immunoblotting: rabbit polyclonal anti-claudin2 (1:250) (#51-6100 Invitrogen, Thermo-Fisher Scientific, Waltham, MA, USA); rabbit polyclonal LC3A/B (D3U4C) (1:1000) (#12741 Cell Signaling Technology); rabbit polyclonal p62 (1:1000) (#39749 Cell Signaling Technology); and rabbit polyclonal anti-β-actin (1:1500) (A2066 Sigma-Aldrich, St. Louis, MI, USA). Immunoreactive proteins were revealed by enhanced chemiluminescence (ECL) and semi-quantitatively estimated using an LAS800 Image Station. Normalization in the same sample was carried out with respect to the β-actin homogenate samples [52,53].

### 3.7. Real-Time Quantitative PCR (qPCR)

For qPCR assays, total RNA was extracted using the RNeasy Mini kit (Qiagen, Hilden, Germany), according to the manufacturer’s instructions. The RNA concentration was determined spectrophotometrically at 260 nm. RNA (2 µg) was retrotranscribed into cDNA using the SuperScript VILO cDNA Synthesis kit (Invitrogen) at the following conditions: 10 min at 25 °C and 60 min at 42 °C. The reaction was terminated at 85 °C for 5 min and cDNAs were stored at −20 °C. For each target, 50 ng cDNA from total RNA were amplified in triplicate in an ABI Prism 7500 HTSequence Detection System (Applied Biosystems, Waltham, MA, USA). 5× HOT FIREPol^®^ EvaGreen^®^ qPCR Mix Plus (ROX) (Solis BioDyne, Tartu, Estonia) was used at the following conditions: 95 °C for 15 min, and 40 cycles of 95 °C for 15 s, 62.5 °C for 20 s, and 72 °C for 20 s. The sequences of the primers used (Sigma-Aldrich, St. Louis, MI, USA) are claudin-2: F-TCCCCAAACCCACTAATCACA; R-CCAACCTCAGCCAGAGAGAGG; LC3: F-CAGCATCCAACCAAAATCCC; R-GTTGACATGGTCAGGTACAAG; p62: F-CCAGAGAGTTCCAGCACAGA; R-CCGACTCCATCTGTTCCTCA; β-actin: F-TGTGGCATCCACGAAACTAC; R-GGAGCAATGATCTTGATCTTCA. For the relative quantification of each target vs. β-actin mRNA, the comparative CT method was used.

### 3.8. Statistical Analysis

All data are expressed as mean ± SEM (standard error of the means). The values were compared to the negative control (untreated cells) or positive control (inflammatory stimulus) using the Tukey test following one-way ANOVA calculation. A *p*-value < 0.05 was considered to be statistically significant.

## 4. Conclusions

Taken together, our studies show that an inflammatory stimulus, such as TNF-α/IL-1β cytokine administration, induces TEER alterations related to claudin-2 expression in a biphasic modulation during 24 h of treatment. Elevated claudin-2 protein levels after subsequently repeated stimulations could trigger a wider effect on intestinal barrier permeability change. In this scenario, pre-treatment with cinnamon extract seems to prevent TEER loss and claudin-2 overexpression, simultaneously influencing mRNA transcription and protein degradation via the autophagy machinery. Various polyphenols in the extract might act synergistically on the different pathways analysed. The regulation of autophagy functional status by bioactive compounds may prevent or delay chronic disease progression, and therefore the improvement in research on autophagy-associated functional food development is becoming mandatory. Cinnamon extract, used as an ingredient in functional foods, may thus be incorporated into an efficient nutritional approach to prevent relapse in IBD or other chronic inflammatory pathologies.

## Figures and Tables

**Figure 1 ijms-24-09201-f001:**
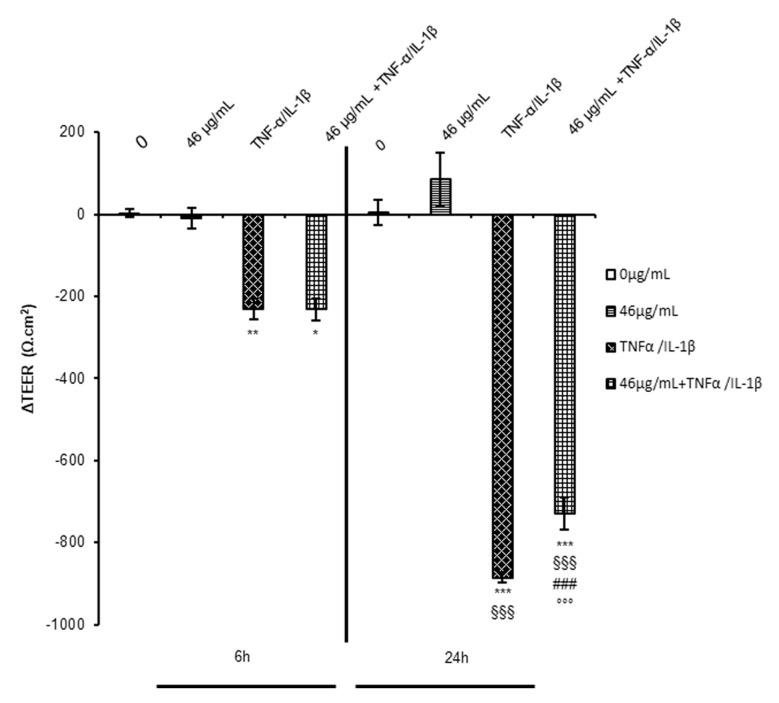
Measurement of TEER under different experimental conditions. Caco-2 cells, cultured on Transwell inserts for 21 days were exposed to pro-inflammatory cytokines (TNFα 10 ng/mL + IL-1β 5 ng/mL) for 6 or 24 h after a 24 h pre-treatment with or without digested cinnamon extract (46 μg/mL in polyphenols). TEER measurements (Ώ.cm^2^) were performed before (0 h) and after (6 or 24 h) cytokine treatment. The ΔTEER of cells exposed to cytokines was calculated and reported in the histograms as numbers. The data represent the mean ± SEM from at least three independent experiments. Statistical significance: * *p* < 0.05, ** *p* < 0.01, and *** *p* < 0.001 vs. untreated cells (6 h); ^§§§^ *p* < 0.001 vs. TNFα/IL-1β (6 h); ^###^ *p* < 0.001 vs. 46 μg/mL + TNFα/IL-1β (6 h); and °°° *p* < 0.001 vs. TNFα/IL-1β (24 h).

**Figure 2 ijms-24-09201-f002:**
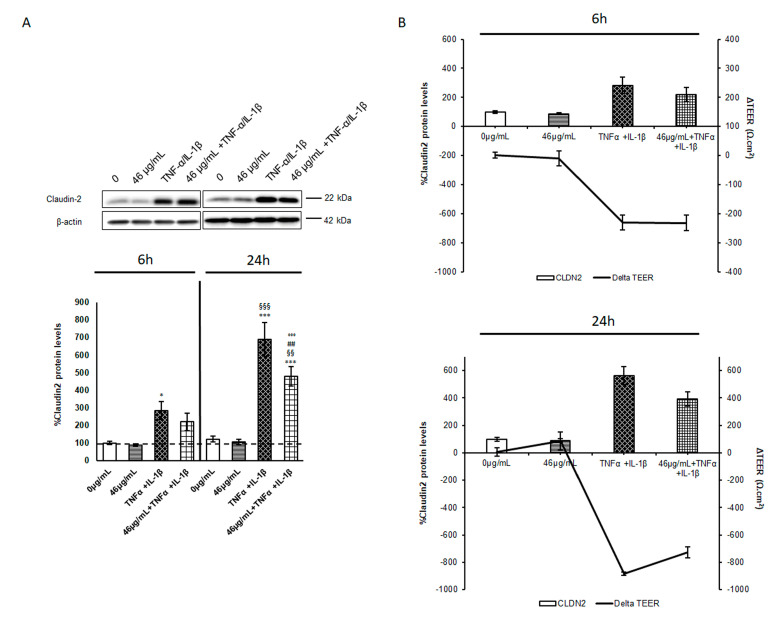
Evaluation of claudin-2 protein levels. Cell lysates were harvested after different treatments, and samples analysed for protein concentration by bicinchoninic acid (BCA) assay. Equal amounts of homogenate samples (as 25 µg protein) were analysed by SDS-PAGE electrophoresis and Western blotting. Claudin-2 was detected with a specific antibody and revealed by enhanced chemiluminescence (ECL). Samples were normalized on β-actin immunoreactivity. (**A**) Histograms, obtained from at least three distinct experiments, represent the percentage of protein levels with respect to untreated cells (6 h) as mean ± S.E. Statistical significance: * *p* < 0.05 and *** *p* < 0.001 vs. untreated cells (6 h); ^§§^ *p* < 0.01 and ^§§§^ *p* < 0.001 vs. TNFα/IL-1β (6 h); ^##^ *p* < 0.01 vs. 46 μg/mL + TNFα/IL-1β (6 h); and °°° *p* < 0.001 vs. TNFα/IL-1β (24 h). (**B**) Claudin-2 protein levels at the two different time points were correlated to TEER fluctuations and reported in the graphic as histograms for claudin 2 (left *y*-axes) and lines for ΔTEER (right *y*-axes).

**Figure 3 ijms-24-09201-f003:**
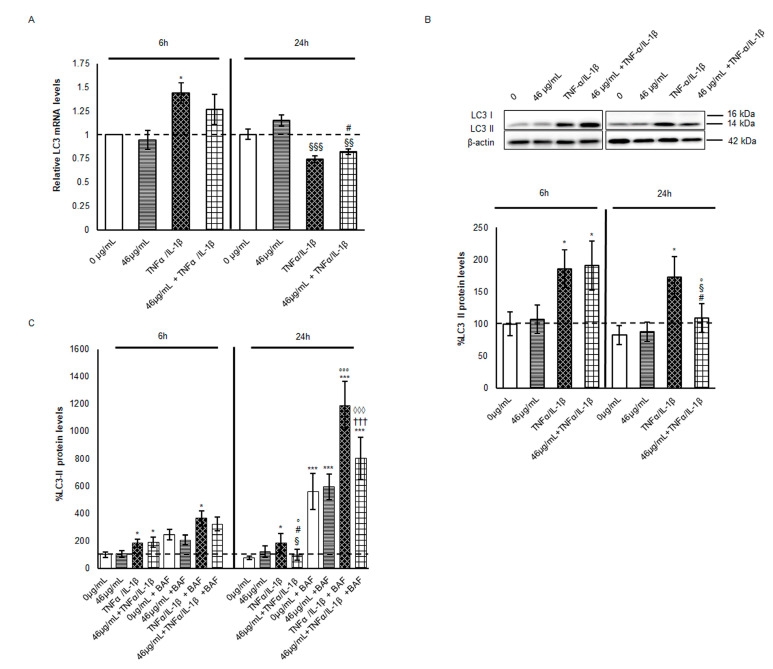
Evaluation of autophagy marker LC3 mRNA and protein levels. (**A**) mRNA was extracted by cells harvested after different treatments. Equal amounts of retrotranscribed cDNA were amplified by RT-PCR. For the relative quantification of each target vs. β-actin mRNA, the comparative C T method was used. Histograms, obtained from at least three distinct experiments, represent the percentage of protein levels with respect to untreated cells (6 h) as mean ± S.E.M. (**B**,**C**) Cell lysates were harvested after the different treatments. In (**C**), BafA1 was administered for 4 h or 18 h at a concentration of 100 nM to Caco-2 cells during cytokine exposure. Equal amounts of homogenate samples (as 25 µg protein) were analysed by SDS-PAGE electrophoresis and Western blotting. LC3 was detected with a specific antibody and revealed by enhanced chemiluminescence (ECL). The samples were normalized on β-actin immunoreactivity. Histograms, obtained from at least three distinct experiments, represent the percentage of protein levels with respect to untreated cells (6 h) as mean ± S.E. Statistical significance: * *p* < 0.05 and *** *p* < 0.001 vs. untreated cells (6 h); ^§^ *p*< 0.05, ^§§^ *p* < 0.01 and ^§§§^ *p* < 0.001 vs. TNFα/IL-1β (6 h); ^#^ *p* < 0.05 vs. 46 μg/mL + TNFα/IL-1β (6 h); ° *p* < 0.05 and °°° *p* < 0.001 vs. TNFα/IL-1β (24 h); ^†††^ vs. 46 μg/mL+ TNFα/IL-1β (24 h); and ^◊◊◊^ *p* < 0.001 vs. TNFα/IL-1β + BAF 24 h.

**Figure 4 ijms-24-09201-f004:**
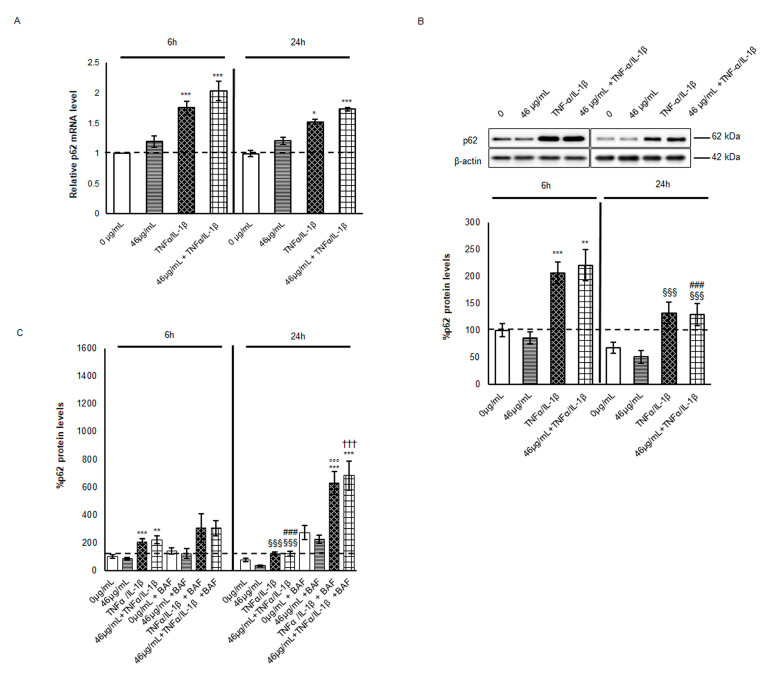
Evaluation of autophagy marker p62 mRNA and protein levels. (**A**) mRNA was extracted by cells harvested after different treatments. Equal amounts of retrotranscribed cDNA were amplified by RT-PCR. For the relative quantification of each target vs. β-actin mRNA, the comparative CT method was used. Histograms, obtained from at least three distinct experiments, represent the percentage of protein levels with respect to untreated cells (6 h) as mean ± S.E.M. (**B**,**C**) Cell lysates were harvested after the different treatments. In (**C**), BafA1 was administered for 4 h or 18 h at a concentration of 100 nM to Caco-2 cells during cytokine exposure. Equal amounts of homogenate samples (as 25 µg protein) were analysed by SDS-PAGE electrophoresis and Western blotting. p62 was detected with a specific antibody and revealed by enhanced chemiluminescence (ECL). The samples were normalized on β-actin immunoreactivity. Histograms, obtained from at least three distinct experiments, represent the percentage of protein levels with respect to untreated cells (6 h) as mean ± S.E. Statistical significance: * *p* < 0.05, ** *p* < 0.01 and *** *p* < 0.001 vs. untreated cells (6 h); ^§§§^ *p* < 0.001 vs. TNFα/IL-1β (6 h); ^###^ *p* < 0.001 vs. 46 μg/mL + TNFα/IL-1β (6 h); °°° *p* < 0.001 vs. TNFα/IL-1β (24 h); and ^†††^ vs. 46 μg/mL + TNFα/IL-1β (24 h).

**Figure 5 ijms-24-09201-f005:**
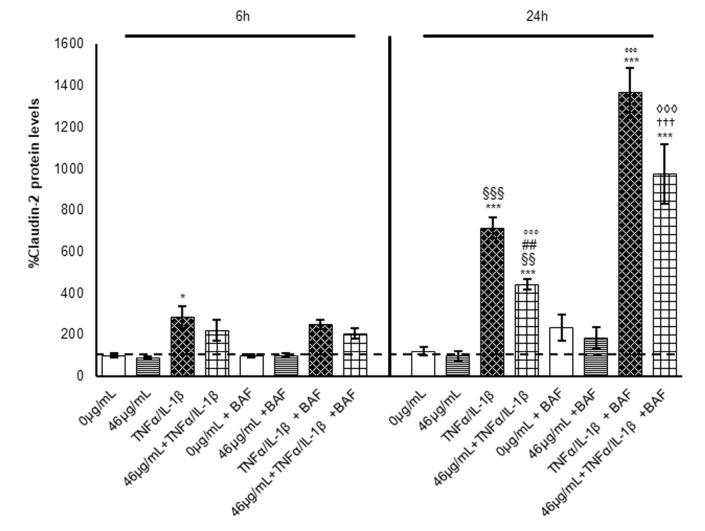
Evaluation of claudin-2 protein levels under autophagy block. BafA1 was administered for 4 h or 18 h at a concentration of 100 nM to Caco-2 cells during cytokine exposure. Equal amounts of homogenate samples (as 25 µg protein) were analysed by SDS-PAGE electrophoresis and Western blotting. Claudin-2 was detected with a specific antibody and revealed by enhanced chemiluminescence (ECL). The samples were normalized on β-actin immunoreactivity. Histograms, obtained from at least three distinct experiments, represent the percentage of protein levels with respect to untreated cells (6 h) as mean ± S.E. Statistical significance: * *p* < 0.05 and *** *p* < 0.001 vs. untreated cells (6 h); ^§§^ *p*< 0.01 and ^§§§^ *p* < 0.001 vs. TNFα/IL-1β (6 h); ^##^ *p* < 0.01 vs. 46 μg/mL + TNFα/IL-1β (6 h); °°° *p* < 0.001 vs. TNFα/IL-1β (24 h); ^†††^ vs. 46 μg/mL + TNFα/IL-1β (24 h); and ^◊◊◊^ *p* < 0.001 vs. TNFα/IL-1β +BAF 24 h.

**Figure 6 ijms-24-09201-f006:**
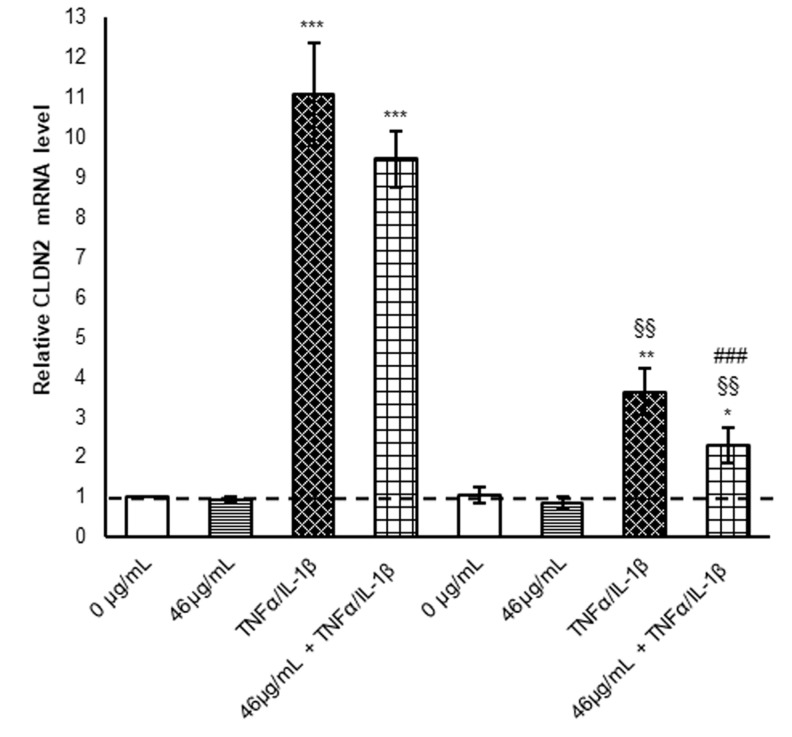
Evaluation of claudin-2 mRNA synthesis. mRNA was extracted by cells harvested after different treatments. Equal amounts of retrotranscribed cDNA were amplified by RT-PCR. For the relative quantification of each target vs. β-actin mRNA, the comparative CT method was used. Histograms, obtained from at least three distinct experiments, represent the percentage of protein levels with respect to untreated cells (6 h) as mean ± S.E.M. Statistical significance: * *p* < 0.05, ** *p* < 0.01 and *** *p* < 0.001 vs. untreated cells (6 h); ^§§^ *p* < 0.01 vs. TNFα/IL-1β (6 h); and ^###^ *p* < 0.001 vs. 46 μg/mL + TNFα/IL-1β (6 h).

## Data Availability

All related data and methods are presented in this paper. Additional inquiries should be addressed to the corresponding author.

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
