# Peer review of "Digested Cinnamon (Cinnamomum verum J. Presl) Bark Extract Modulates Claudin-2 Gene Expression and Protein Levels under TNFα/IL-1β Inflammatory Stimulus"

_ijms, 2023, doi:10.3390/ijms24119201_

Round 1
Reviewer 1 Report
The manuscript presents interesting data indicating that the Cinnamon verum aqueous extract, subjected to a process of gastrointestinal digestion simulation, exerts beneficial anti-inflammatory properties in an in vitro model of inflamed intestinal barrier. The treatment with cinnamon extract prevented TEER loss; moreover, a correlation between TEER decrement and alterations in Claudin-2 expression were found. These data are interesting from the point of view of the prevention of progression of the inflammatory bowel disease.
Remarks:
In the Abstract and in the Introduction, the Authors mention antioxidant properties of the extract studied. Can these antioxidant properties participate in the effect observed? (To be perhaps mentioned in the Discussion).
Line 25: “TEER”, please explain the acronym
Line 82: “recently grown”, either “grew” or “has grown”
Lines 114 and 437: Please give “2” in a superscript
Figure 2A: Is the blot shown representative for the plot? On the blot, it seems that the amount of Clauin-2 is significantly lower for the treatment with 46 μg/mL extract treatment than in the control.
Line 185: What amounts?
Lines 372-374: tap water or distilled (deionized) water?
Line 442: “12,5%”, please change to “12.5%”
I detected one problem (listed; "grown" used inapproprietely but perhaps there are some other small problems which escaped my attentiion. Generally, the English seems good to me,
Reviewer 2 Report
The article entitled " Cinnamon (Cinnamomum verum J. Presl) bark digested extract modulates Claudin-2 gene expression and protein levels under TNFα/IL-1β inflammatory stimulus." presented to me for review is original work. The authors try to prove the that cinnamon pre-treatment plays a positive role in regulating Claudin-2 levels in relation to TEER values, through a fine modulation of both Claudin-2 autophagy-mediated degradation and gene transcription. In my opinion, the article is interesting but somewhat similar to the earlier article presented by the authors (doi: 10.3390/foods12030452). The abstract and the Introduction are written correctly but the introduction lacks a specific purpose of the work, which should be added at the end - please make it clear. The results section are presented in a correct way, but the graphs and gel pictures are in very poor resolution and it is difficult to deduce anything from them - please correct. In addition, there is no discussion section, although it is probably a mistake because in the results section it can see some signs of discussion, although it is presented in a rather laconic way. The lack of specific examples slightly reduces the substantive value of this article, so please add to it. the conclusions are too long and I would rather avoid citations in this section. They should include a concise summary of your most important achievements. Besides, the authors should explain more specifically on what basis they chose these particular genes and why only two. I also believe that the authors slightly exaggerate the mechanism of action after the expression of two genes. Apart from that, much of the methodology is quite similar to earlier research, and in fact there is little that is new that clearly distinguishes this work. I would suggest the authors to expand the spectrum of research and then draw some specific conclusions. Please also check and correct any grammatical and stylistic errors in the article.
Minor editing of English language required.
